# Weakening of resistance force by cell–ECM interactions regulate cell migration directionality and pattern formation

Masaya Hagiwara [1,2✉], Hisataka Maruyama[3], Masakazu Akiyama[4], Isabel Koh [1] & Fumihito Arai[3,5]

Collective migration of epithelial cells is a fundamental process in multicellular pattern formation. As they expand their territory, cells are exposed to various physical forces generated by cell–cell interactions and the surrounding microenvironment. While the physical stress applied by neighbouring cells has been well studied, little is known about how the niches that surround cells are spatio-temporally remodelled to regulate collective cell migration and pattern formation. Here, we analysed how the spatio-temporally remodelled extracellular matrix (ECM) alters the resistance force exerted on cells so that the cells can expand their territory. Multiple microfabrication techniques, optical tweezers, as well as mathematical models were employed to prove the simultaneous construction and breakage of ECM during cellular movement, and to show that this modification of the surrounding environment can guide cellular movement. Furthermore, by artificially remodelling the microenvironment, we showed that the directionality of collective cell migration, as well as the three-dimensional branch pattern formation of lung epithelial cells, can be controlled. Our results thus confirm that active remodelling of cellular microenvironment modulates the physical forces exerted on cells by the ECM, which contributes to the directionality of collective cell migration and consequently, pattern formation.

[1] Human Biomimetic System RIKEN Hakubi Research Team, RIKEN Cluster for Pioneering Research (CPR), Wako, Saitama, Japan. [2] Department of Biological System, Osaka Prefecture University, Sakai, Osaka, Japan. [3] Department of Micro-Nano Mechanical Science and Engineering, Nagoya University, Nagoya, Japan. [4] Institute for Advanced Study of Mathematical Science, Meiji University, Tokyo, Japan. [5] Department of Mechanical Engineering, The University of Tokyo, Tokyo, Japan. ✉email: masaya.hagiwara@riken.jp

Collective cell migration refers to the coordinated movement of groups of cells in response to interactions with the surrounding microenvironment and other cells. It is a fundamental process in multicellular pattern formation[1–3], wound healing[4,5], and tumour invasion[6,7]. In particular, collective epithelial dynamics contribute greatly to the formation of tissues with complex patterns, such as the lung[8–10], mammary gland[11], salivary gland[12], and kidney[13,14] via various morphological events that occur within epithelial cell sheets including invagination and cellular outgrowth into the extracellular matrix (ECM)[15,16]. The mechanisms underlying this pattern formation have been well studied experimentally and theoretically in terms of physical cell–cell interactions[17–21] or coordinated chemical systems[10,15,22]. In contrast, little is known about the contribution of ECM remodelling and cell–ECM interactions to the directionality of collective epithelial cell migration. Cells move dynamically in the surrounding ECM, which is an essential scaffold composed of proteins such as collagen that supplies chemical and mechanical cues for tissue morphogenesis[23], by degrading the ECM to increase the pore size of the scaffold and alter adhesion[24,25]. In addition, cells also produce ECM proteins, such as fibronectin (FN), which contribute to cell rearrangement by forming cell adhesion sites[26,27]. These remodelling events in the cellular microenvironment are known to modulate the forces exerted on cells and affect individual cell migration in fibroblasts and mesenchymal cells[25,28,29] as well as in endothelial cells during vascularisation and angiogenesis[30–33]. However, the mechanism via which spatio-temporal remodelling of the microenvironment contributes to large multicellular pattern formation remains unclear; understanding the role of ECM remodelling on the directionality of collective epithelial cell migration within an epithelial sheet is essential to understand the mechanisms of pattern formation, wound healing, and tumour invasion.

On the other hand, a variety of engineering technologies, such as microfabrication and microrobots driven by optical or magnetic forces, have been developed to control cellular microenvironment, and to manipulate small objects. For example, the precise control of cell seeding position and geometry by using photolithography methods[34] leads to high repeatability of the direction of collective cell migration[35,36], enables quantitative analysis of cellular behaviour[37,38], and can be used to direct cell differentiation[39,40]. Optical tweezers are among the major systems that are used to manipulate micro-objects surrounding cells with sub-nanometre spatial resolutions[41–43]. Alternatively, magnetic force is another major method for the manipulation of micro-objects[44,45], by exerting forces in the order of millinewtons on the object. Hence, these robots can be used to conduct force-requiring tasks, such as the cutting of cells. By employing or further developing the above-mentioned engineering technologies, we can analyse the dynamic modulation of ECM during collective cell migration.

Here, we report how remodelling of the ECM governs the directionality of collective epithelial cell migration. The conditions of the cellular microenvironment are constantly being altered by the interaction between cells and the ECM, via the degradation of ECM and the deposition of adhesion sites (Fig. 1a). Thus, we investigated the dynamic modification of ECM by the physical movement of cells, as well as the deposition of FN by cells during cellular migration. Various engineering techniques were employed to analyse the dynamic modification of ECM and its effect on cellular behaviour. Photolithography-based microfabrication enables us to control initial collective cell geometry, and to pattern an FN site to an arbitrary shape on a dish, and an optical tweezer system, which enables non-contact manipulation of microbeads to simulate cellular movement in the ECM, was employed to quantitatively measure the resistance force from the ECM acting on the cells during cellular movement. Normal human bronchial epithelial (NHBE) cells, which form a branched structure when cultured in matrigel[46,47], were used in this study to investigate the role of cell–ECM interaction during pattern formation, and artificial modification of the ECM in 2D and 3D demonstrated that the modulation of the microenvironment directs collective cell migration. Thus, this study shows the importance of designing the cellular environment in order to reconstitute tissue pattern formation.

## Results

**Regulation of cellular directionality by construction of an FN site.** FN is widely expressed by various types of cells, and contributes to the interaction between cells and the ECM by supplying cells with strong adhesion sites, as well as stimulating the mechanosensors on cell surfaces; the integrin-based focal adhesion sites on cells sample ECM rigidity via mechanosensors, which guide directed migration[48–50]. Thus, it is expected that a fabricated FN site would also regulate cell migration. However, it is difficult to analyse the effects of FN on cellular behaviour if the cell is randomly walking in a low-repeatable manner. Control of the initial cell culture conditions, such as collective cell geometry and density, provides great advantages in terms of the repeatability of cellular behaviour and pattern formation. Therefore, we have employed multiple microfabrication technique to analyse the function of FN on cellular behaviour. A poly-dimethylsiloxane (PDMS) membrane with a triangular-shaped hole was fabricated by photolithography and placed on a dish, then NHBE cells were seeded over the PDMS membrane. After the cells have adhered to the dish, the PDMS was removed. When NHBE cells were seeded on a triangular-shaped pattern on a dish and covered by matrigel, cells initially migrated toward the tip area, leaving the centre of the triangle empty. As discussed in our previous paper[35], this is thought to be mainly due to the chemotaxis associated with the reaction-diffusion system. Then, cells start to meander inside the triangle. Figure 1b shows the FN immunostaining results after 24 hours of culturing. The cells were observed to aggregate in the lower right corner, and there were no cells present in the upper left area. However, a cellular footprint had been generated, and FN remained at the initial cell seeding site, even at sites where cells were no longer present.

In order to examine how the deposited FN site affects cellular behaviour, a dual patterning method was established to enable both collective cell patterning and FN coating with arbitrary shapes on a dish (Fig. 1c). By employing this technique, the cell culture area and geometry could be controlled, and at the same time, the area surrounding the cells can be patterned with FN. NHBE cells were once again cultured in the middle of the FN-patterned site (Fig. 1d), and the boundary of FN site could be clarified clearly, allowing the analysis of cell behaviour at the edge of the site. Time-lapse images were taken to track cellular behaviour around the FN boundary. We first analysed individual cellular behaviour near the FN boundary area by culturing cells at a low density. Cells initially migrated outward from the cell-seeded area, but when the cells reached the boundary of the FN pattern, they turned back. After 40 hours, none of the cells had crossed the boundary (Fig. 1e and Supplementary Movie 1). This result indicated that the cells preferred to stay within the FN site. In contrast, when cell density in the culture area is high, some of the cells crossed the FN boundary by squeezing through the neighbouring cells, and then proceeded to construct new FN site by themselves, while other cells at the boundary moved in parallel to the boundary line to avoid crossing it (Fig. 1f and Supplementary Movie 2). As a control, a PDMS sheet without FN immersion was placed on a dish and the profile of the sheet

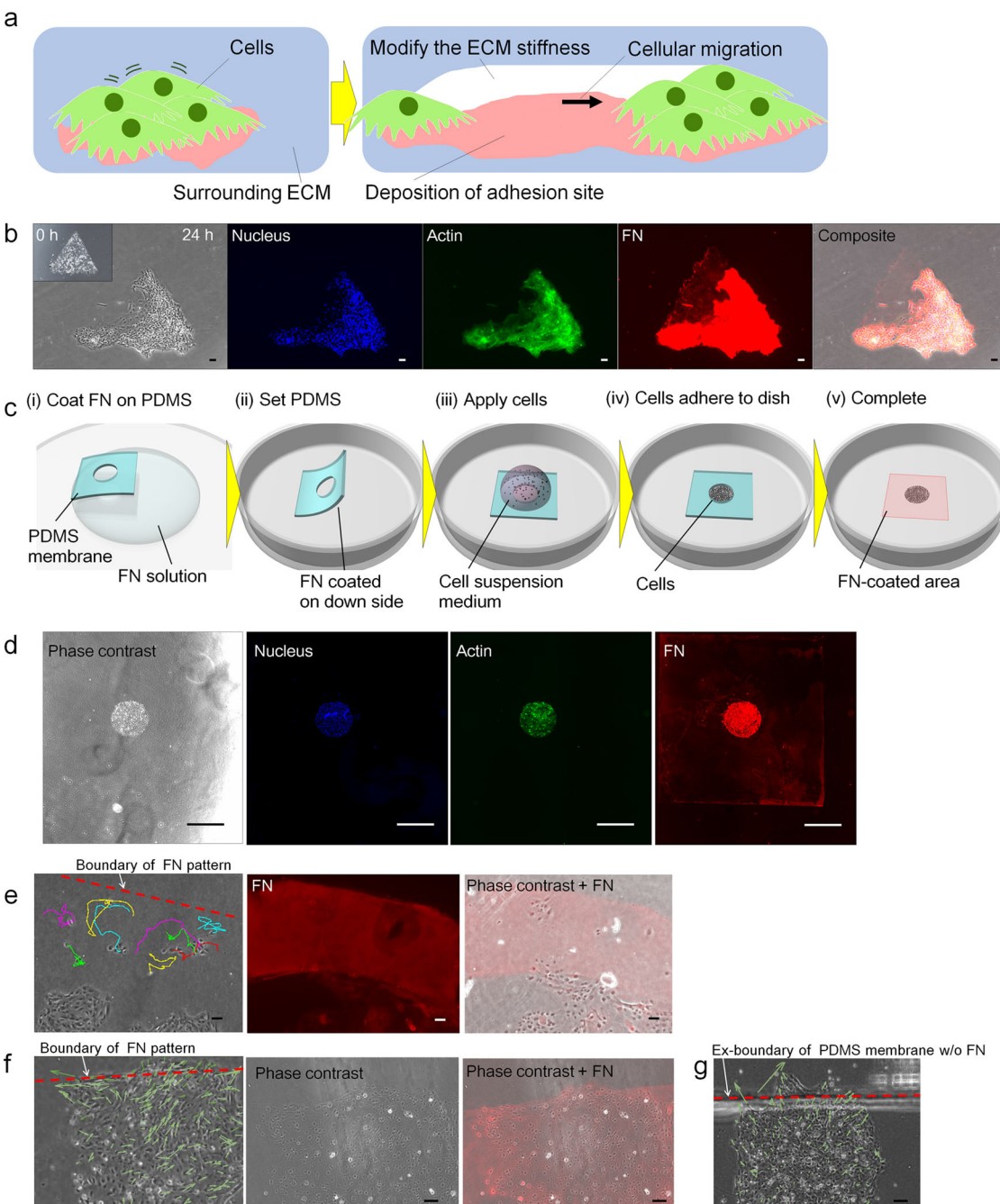

**Fig. 1 Deposition of FN regulates the cellular migration area. a** Schematic representation of microenvironment remodelling. During collective cell migration, ECM is continuously degraded and constructed. **b** Immunostaining results of the nucleus, actin and FN. NHBE cells were initially placed in a triangular shape, and were observed to have migrated towards the lower right corner after 24 hours, although FN remained in the area without cells. **c** Schematics of the procedure for simultaneous dual patterning of cells and FN. **d** Phase contrast and immunostaining of the nucleus (DAPI), actin and FN show the dual patterning of NHBE cells and FN. A rectangular FN boundary was patterned on a dish, and NHBE cells then patterned in the centre of the FN area. **e** Tracking of cell migration inside the FN-patterned area. The red dotted line indicates the FN boundary, which was determined from the staining results. **f** Cellular behaviour in the crowded area near the FN boundary. Immunostaining of FN is shown in red. **g** Control experiment without FN patterning. The complete time-lapse movies for parts **e**–**g** are shown in Supplementary Movies 1–3. Scale bar: 100 μm (**b**, **e**–**g**) and 1 mm (**d**).

boundary was marked under the dish. In this case, cells easily crossed the boundary, unlike in the case with FN patterning (Fig. 1g and Supplementary Movie 3). These results indicate that the construction of an FN site restricts cells to remain in their original sites, but once they are able to break into unexplored areas aided by external forces, such as by being pushed by neighbouring cells, they can expand their territory by depositing FN, and other cells can soon follow them.

**Existence of a resistance force on the cell–ECM interface.** We have previously succeeded in developing collective cell pattern formation in vitro by layering matrigel onto NHBE cells, whose initial collective cell geometry was controlled by photolithography to enhance the repeatability of experiments[35]. This in vitro experimental model can be used to analyse the effects of the surrounding ECM on collective cell migration, and to determine how cells modulate the surrounding ECM to organise pattern

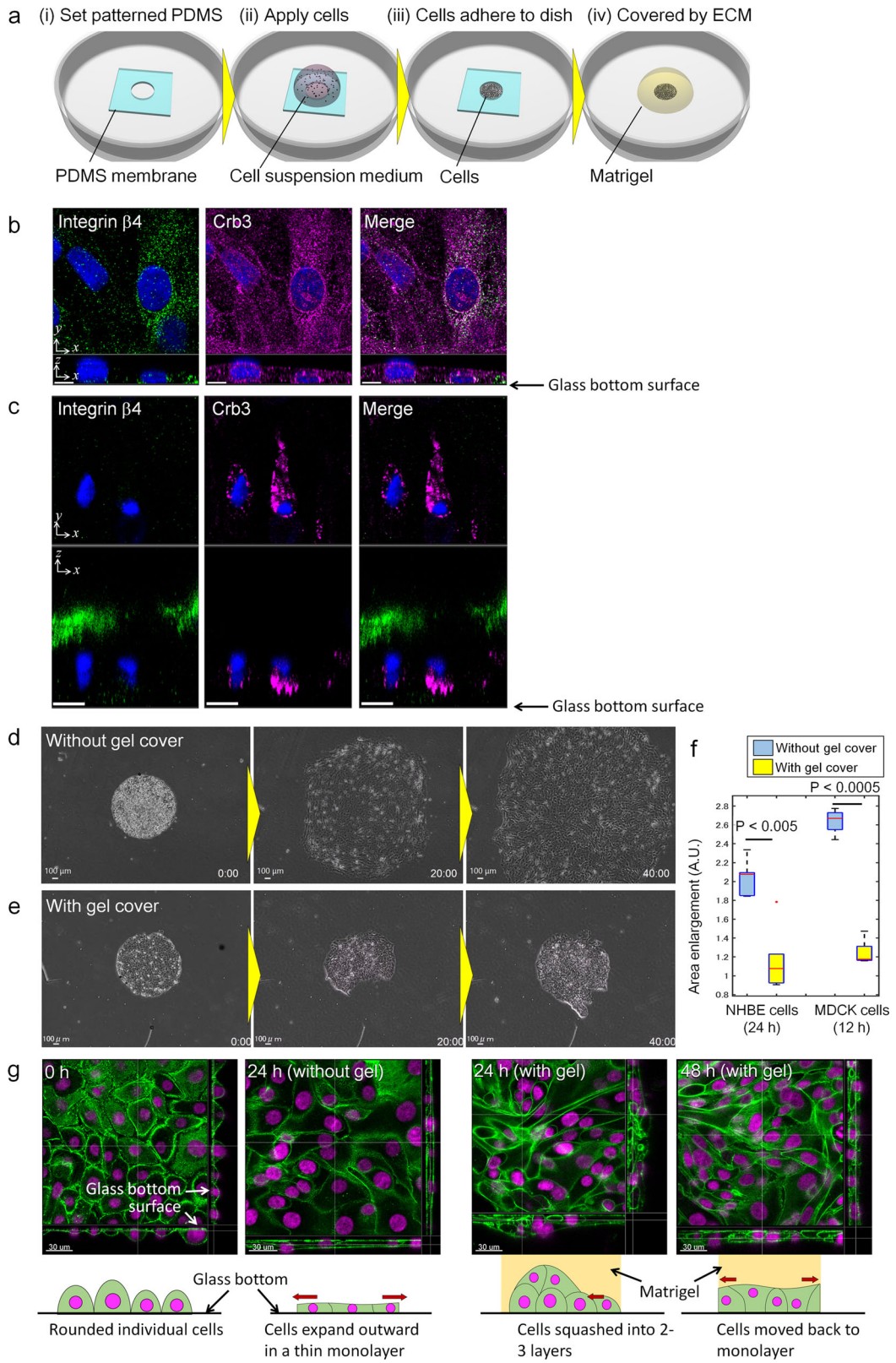

formation. In this paper, we employed this experimental model to investigate in vitro how the interface boundary between cells and the ECM changes dynamically during collective cell migration. To analyse the effects of the surrounding ECM on collective cell migration, NHBE cells were patterned on a plastic dish in an arbitrary shape and covered with growth factor-reduced matrigel (Fig. 2a). In order to clarify the epithelial polarity with and

without matrigel cover, we stained NHBE cells, after 3 days of culture, with integrin β4 and Crb3, which are expressed on the basal and apical sides of cells[51], respectively. When NHBE cells were seeded on a glass-bottom dish without matrigel cover, integrin β4 was expressed on the side towards the glass-bottom surface, whereas Crb3 was distributed homogeneously throughout the cell membrane (Fig. 2b). On the other hand, in NHBE

**Fig. 2 Existence of resistance force between cell–ECM boundary. a** Schematics of the procedure for in vitro migration assay with cell patterning. **b** Integrin β4 and Crb3 immunostaining result of NHBE cells without matrigel cover. **c** Integrin β4 and Crb3 immunostaining result of NHBE cells with matrigel cover. **d** Phase contrast time-lapse images of NHBE cells patterned in a circular shape without matrigel. **e** Phase contrast time-lapse images of NHBE cells patterned in a circular shape with matrigel cover. **f** Comparison of area expansion rates with and without matrigel, for NHBE and MDCK cells. (centre red line, median; box limits, upper and lower quartiles; whiskers, 1.5× interquartile range; points, outliers). **g** z-stack images to visualise the cellular organisation after NHBE cell seeding, at 24 hours of culture with and without matrigel cover, and at 48 hours of culture with matrigel cover. (green: actin, magenta: nucleus). Five culture experiments were conducted for NHBE and MDCK, respectively, and the data include the means ± SDs ($n = 5$) and Student's $t$ test. Scale bar: 10 μm (**b**, **c**), 100 μm (**d**, **e**). The complete time-lapse movies for parts **d** and **e** are shown in Supplementary Movies 4 and 5 for NHBE and in Supplementary Movies 8 for the 3D voxel images of z-stack in part **g**.

cells cultured with a matrigel cover, integrin β4 was more highly expressed on the matrigel side while Crb3 was expressed relatively on the lower side of the cells (Fig. 2c). Comparing the result with and without matrigel cover, cellular polarity was much more distinguished in the case with matrigel cover.

Figure 2d shows the behaviour of NHBE cells that were set in a circular pattern without matrigel, as the control experiment. Without matrigel, the cells expanded their territory beyond the original seeded area while maintaining a rounded collective cell shape (Supplementary Movie 4). On the other hand, the results obtained with matrigel in Fig. 2e showed that NHBE cells initially migrated to the edge of the original collective geometry; this is also owing to the chemotaxis shown in the triangular shape caused by morphogen distribution of short-range activator and long-range inhibitor[35]. After several hours, they are pushed back to the inner regions due to the mechanical interactions of squeezed cells at the edge, and subsequently continued to meander inside the original circular geometry. None of the cells migrated out of the initial geometry within the first 24 hours (Supplementary Movie 5). This tendency of cellular behaviour did not depend on the initial collective cell geometry, with the cells consistently returning to the original geometry after a certain period of time as if they had a memory (Supplementary Fig. 1a, and Movie 6). We also examined the differences in cellular behaviour with and without a matrigel cover using another epithelial cell, Madin-Darby canine kidney (MDCK) cells, and the same tendency of cell expansion was observed; the cells without a matrigel cover migrated outward significantly faster than cells with a matrigel cover (Supplementary Fig. 1b, c, Movie 7). The expansion rates of the collective NHBE cells after 24 hours and those of the MDCK cells after 12 hours, with and without the matrigel cover, were measured. In both cases, the cells lacking a gel cover migrated at a significantly higher speed than those covered with matrigel, which barely migrated further than the initial cell area (Fig. 2f).

In order to analyse cellular organisation with and without matrigel cover, z-stack images were taken after NHBE cells were seeded in a circular pattern, at 24 hours with and without matrigel cover, and at 48 hours with matrigel cover (Fig. 2g and Supplementary Movie 8). Initially, cells still retained their individual, rounded shape just after seeding, but are in contact with each other on the glass-bottom surface of the dish. After 24 hours of culture without matrigel cover, as cells expand outward as shown in Fig. 2d, they flatten into a thin monolayer. On the other hand, after 24 hours of culture with matrigel cover, as cells aggregated inside the original circular pattern as shown in Fig. 2e, the cells were observed to accumulate into two to three layers. This modification of cellular organisation is thought to be owing to cells being squashed by the surrounding matrigel as well as being pushed by neighbouring cells. In fact, after 48 hours of culturing with matrigel cover, as cells recovered to the original circular shape and started to migrate outside of the circular shape, cells transitioned back to a monolayer formation.

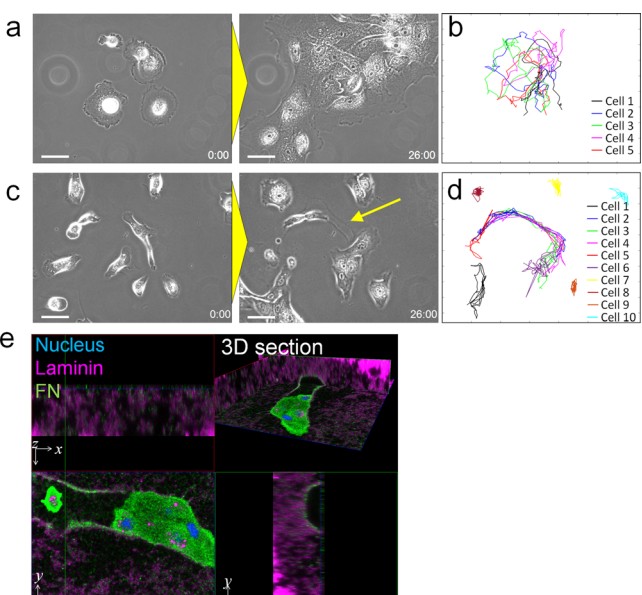

**Fig. 3 Effects of the modification of ECM resistance force on cellular behaviour. a** Time-lapse images of NHBE cell behaviour without matrigel cover. **b** Cell tracking results of cells in Fig. 4a. The initial five cells were tracked over a period of 16 hours. **c** Time-lapse images of NHBE cell behaviour with matrigel cover. The arrow shows the track made by the cell along its path of movement. **d** Cell tracking results of cells in Fig. 4c. The initial 10 cells were tracked over a period of 16 hours. **e** 3D immunostaining of laminin and FN was used to show the distribution of matrigel and FN footprint during cell migration, respectively. A tunnel was visible in the path of cell movement, and traces of FN remained in the tunnel. Scale bar: 100 μm (**a**, **c**, **e**).

These results indicate that a resistance force had been generated at the cell–ECM interface, and that this force prevents cells from migrating out of the initial collective cell geometry. We hypothesise that once the resistance forces exerted on the cells from the ECM are weakened, or once the cellular driving forces exceeds the resistance force from ECM, then cells can expand their territory.

**Carving of ECM by cellular movement.** To further investigate how cellular movement affects the surrounding ECM and the behaviour of other cells, time-lapse imaging was conducted to track the movement of single NHBE cells with and without matrigel cover. Without a matrigel cover, dispersed cells congregated together with neighbouring cells soon after seeding to form a cluster and meandered around collectively (Fig. 3a, b, Supplementary Movie 9). On the other hand, many cells under a matrigel cover were initially restricted in their movements but were gradually able to expand their movable range over time (Fig. 3c, d, Supplementary Movie 10). In addition, cells left a track

in the path in which they had migrated (arrow in Fig. 3c) in the ECM, and other cells tended to follow along this same path rather than explore untrampled areas. As a result, the areas in which cells moved were quite limited. To further investigate this track, the sample was stained with laminin, which is a major composition of matrigel, and FN after the cells had made tracks, and observed using 3D confocal microscopy. As shown in Fig. 3e, a tunnel was clearly observed on the track, and FN had been deposited on the basal (matrigel) side of the cell in the tunnel. The result indicates that cells had carved the surrounding ECM during migration, and in doing so, weakened the resistance force from ECM. Simultaneously, FN was deposited on the track on the ECM, which helps cells to migrate as shown in Fig. 1. Together, cells expand their territory by carving the ECM, whereas other cells can easily migrate along the same path, where the resistance force had been lowered.

**Quantitative measurement of resistance force reduction by physical movement of object inside hydrogel.** The fact that cells could hardly move inside the matrigel at the beginning of culture indicated that the resistance force from matrigel was initially higher than the driving force of cell movement. If it is assumed that a single cell's driving force cannot be significantly increased without additional external forces, it brings up an important question of how cells weaken the resistance force from ECM to make a tunnel over time. Although it is now known that cells dynamically alter their surrounding ECM, there is currently no report discussing the physical weakening of ECM resistance due to cellular movement quantitatively. To examine if the physical movement of cells can weaken the resistance force from the surrounding ECM, we manipulated microbeads, by using optical tweezers, to emulate cellular movement in matrigel and collagen I, respectively, then analysed how the repetitive push-pull movement of 'cells' physically degraded the ECM. In total, 600 mW of laser power with a 1064 nm wavelength was outputted through a ×100 oil objective lens (NA 1.4) (Fig. 4a) to manipulate 10 μm beads suspended in twofold diluted matrigel, or in 0.6 mg ml$^{-1}$ collagen I. The increase in temperature by the 600 mW laser was measured by using rhodamine B as an optical thermometer[52], and was ~9°C in matrigel and 17°C in collagen after the laser was turned on. The temperatures reached up to 46°C in matrigel and 54°C in collagen (Supplementary Fig. 2), which is lower than the transition temperature of collagen[53] and thus, we considered that this increase in temperature does not deteriorate the ECM.

When the laser is close to the beads, the beads are trapped by the laser and move in the direction of laser scan; when the resistance force becomes greater than the maximum trap force of the laser, the beads are released from the laser (Fig. 4b). This cycle was repeated up to 100 times along the $y$ axis, and the range of microbead movement along the $x$ and $y$ axes was measured every 20 cycles (Fig. 4c, d). The results showed that the movable area in both matrigel and collagen increased significantly with increasing cycle number in the direction of laser scanning ($y$), whereas there was only a slight increase or no difference in the $x$ direction. This is because the majority of the forces exerted on the ECM by the microbeads were applied to push the ECM in the $y$ direction, whereas only a small amount of shear stress and traction force was applied in the $x$ direction. These results indicate that ECM modulation occurred locally. The resistance force exerted by both hydrogels against the manipulation of the 10 μm beads was also calculated by measuring the distance from the centre of the laser to the centre of the bead after each repetitive movement of the microbeads (Fig. 4e), and converting the distance into force using theoretical model based on the Stokes's law[54] (Supplementary Fig. 3). The ECM resistance clearly decreased as the number of

cycles increased in both matrigel and collagen, and was ~30% lower than the initial value after 40 iterations (Fig. 4f, g and data are available in Supplementary Data 1). These results confirmed that the ECM resistance could be weakened by repetitive internal movements.

**Theoretical analysis of the cellular directionality governed by dynamic change of resistance force from ECM.** A mathematical model was developed to back up the above hypothesis. Four conditions, cell moving polarity, cell-to-cell adhesion, chemotaxis, and cell-to-ECM interactions were taken to be associated with cellular migration in this model (Fig. 5a). The cell polarity and adhesion model is based on the previous work from Akiyama et al.[55]; cells migrate towards a specific direction based on the front-to-back polarity of cells, and cell-to-cell adhesion forces arise when the distance between individual cells is close enough to exert a jostling force on each cell. Chemotaxis of cells is generated by the spatio-temporal interactions of morphogens and largely contributes to cell directionality and pattern formation. For chemotaxis in NHBE cells, we previously confirmed the good agreement of collective cell pattern formation in 2D[35] and 3D[36] with a mathematical model based on H. Meinhardt's model[56]. We also included in this model our hypothesis that cells are faced with a resistance force from the surrounding ECM, and this resistance force is weakened by repetitive movements of an object in the ECM. Figure 5b shows the simulation result of cell movement without a hydrogel cover. As there is no resistance force from the surrounding environment, the direction of cellular migration is determined by the balance between cell polarity and adhesion to neighbour cells. Cells migrated in a straight manner until they meet other cells. Once they get together and form a cluster, the cell collective started to meander about together, just as in the in vitro results (Fig. 5c and Supplementary Movie 11). On the other hand, in the simulation with a matrigel cover, the area in which cells can move is initially limited, and cells can barely travel far (Fig. 5d). However, repetitive movement of cells in their small local area gradually weakened the resistance force from the surrounding ECM, leading to the surrounding ECM being carved. With an increase in the number of cells owing to proliferation and the formation of cell clusters, the areas with low resistance area are gradually expanded and cells can easily migrate back-and-forth along the same path that other cells had already made (Fig. 5e and Supplementary Movie 12). The simulation results taking our hypothesis into account regarding cell–ECM interactions agrees well with the in vitro results of both cases, with and without matrigel (Fig. 3a, c). Both in vitro and simulation results indicated that cells are continuously remodelling the surrounding ECM by reducing the resistance force that hinders migration, and that this remodelling significantly contributes to the direction of cellular migration.

**Controlling the direction of collective cell migration by artificially remodelled microenvironments.** The results so far have shown that a decrease in the resistance force from the ECM is one of the key factors that determine the direction of cellular migration. In other words, the direction of cellular migration can be controlled by artificial modification of the resistance force in the surrounding niche. To examine how the directionality of collective cell migration can be controlled by altering ECM conditions, a limited area of the matrigel was artificially carved by manual manipulations prior to the culturing of cells. As the effect of chemotaxis is more apparent with a larger number of NHBE cells. We employed the same assay as Fig. 2b to analyse multicellular behaviour rather than single-cell behaviour. Similar to the experiment with optical tweezers, magnetic microbeads (5 μm

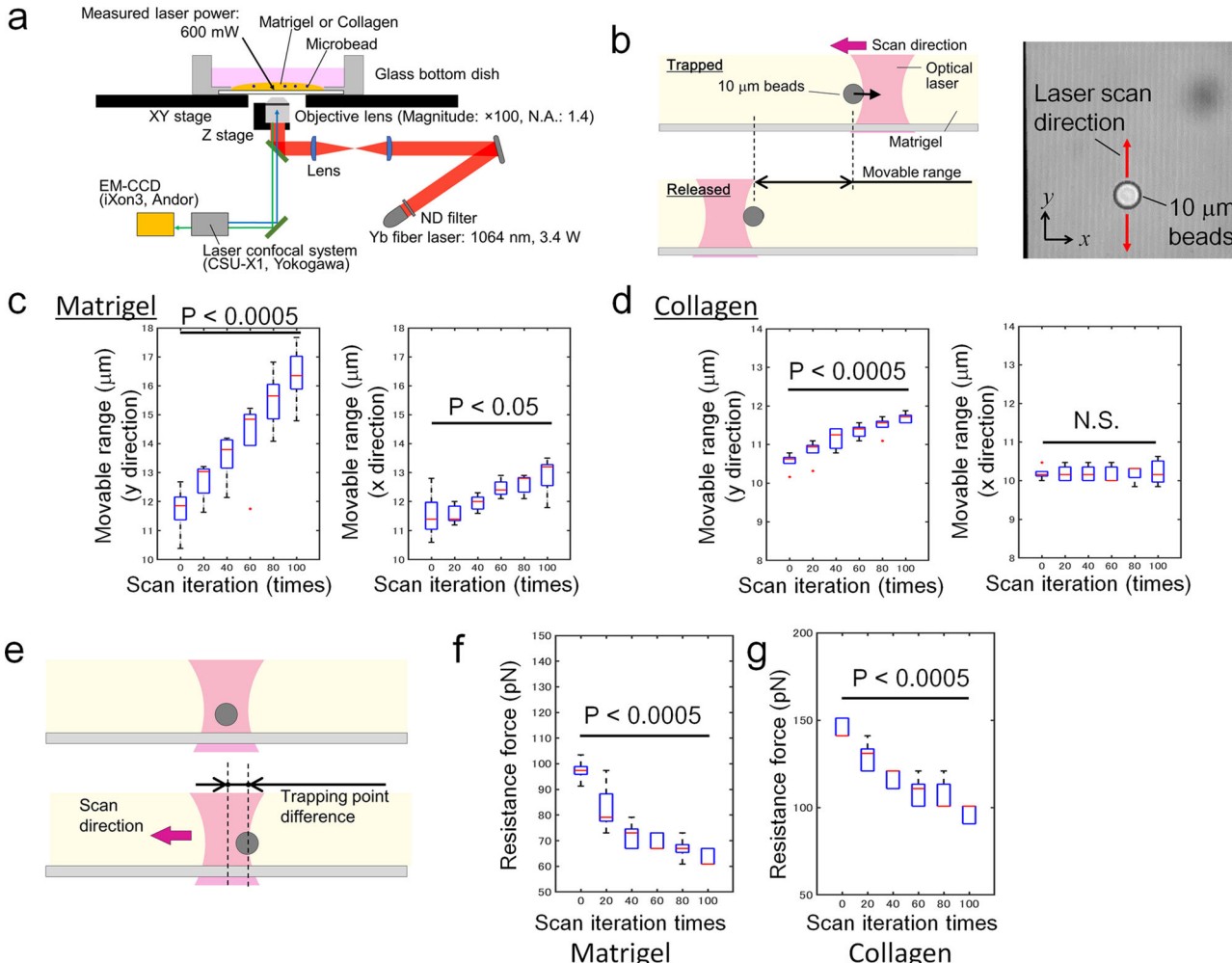

**Fig. 4 Effects of physical movements in the ECM on resistance force. a** Experimental setup for optical tweezers system. **b** Schematic diagram for the measurement of the movable range of microbeads under constant manipulation to examine the modulation of ECM by the repetitive movement of micro-objects. The 10 μm beads were repetitively manipulated along the y axis by optical tweezers with 600 mW of output laser power. **c** Experimental results for the measurement of the movable range of the microbeads in matrigel. Five microbeads in matrigel were selected to conduct the experiments, and the data show the means ± SDs (n = 5). **d** Experimental results for the measurement of the movable range of the microbeads in collagen I. Five microbeads in collagen were selected to conduct the experiments, and the data show the means ± SDs (n = 5). **e** Schematic diagram of the measurement of resistance force from the ECM using optical tweezers. In all, 10 μm beads were repetitively moved, and the dynamic change in resistance force was calculated by measuring the distance between the centre of the laser and that of the trapped beads. **f** Experimental results of the resistance force in matrigel. Five microbeads in matrigel were selected to conduct the experiments, and the data show the means ± SDs (n = 5). **g** Experimental results of the resistance force in collagen I. Five microbeads in collagen were selected to conduct the experiments, and the data show the means ± SDs (n = 5). Student's t test was employed for the statistics (**c, d, f, g**). The elements in boxplots are defined as follows; centre line, median; box limits, upper and lower quartiles; whiskers, 1.5× interquartile range; points, outliers.

diameter) were placed in matrigel to mimic cells, and the matrigel was physically carved by the collective movements of beads driven by a permanent magnet. The controllability of magnetic force is lower than that of optical tweezers; however, this method allows output forces at the milli-newton scale[45], which better mimics the collective cell migration in ECM.

Once the NHBE cells were patterned in a circular shape on a dish, the magnetic beads mixed in matrigel were aligned close to the area of cell accumulation using a neodymium permanent magnet that had a magnetic flux density of 508 mT on the surface. After the gelation of matrigel by incubation at 37 °C for 25 min, the magnetic beads were manipulated with the magnet from under the dish to weaken the resistance force from matrigel via a linear reciprocating motion that resembles collective cell movement. To prevent unknown effects in cellular behaviour, most of the magnetic beads were moved away from the area of

cell accumulation after 10–20 repetitive pushes of the magnet. Upon incubation, the cells initially meandered only within the original site; subsequently, the cells progressively invaded the area carved by the magnetic beads (Fig. 6a and Supplemental Movie 13). Figure 6b shows the collective cell geometry profiles after 52 hours of culture (n = 5). Most of the cells migrated along the axis of the path taken by the magnetic beads and exhibited elongated profiles. Figure 6c, d show the probability of cell presence with respect to the angle from the axis of the path taken by the magnetic beads; ~75% of the cells migrated within ±40 degrees along this axis (Fig. 6c, d). These results indicate that local ECM depression, caused by the movement of the magnetic beads, can direct collective cell migration via a decrease in the resistance force acting on the cells, providing access to a new site that the cells had not previously explored. On the other hand, FN was not present in the area of artificially degraded ECM at the beginning.

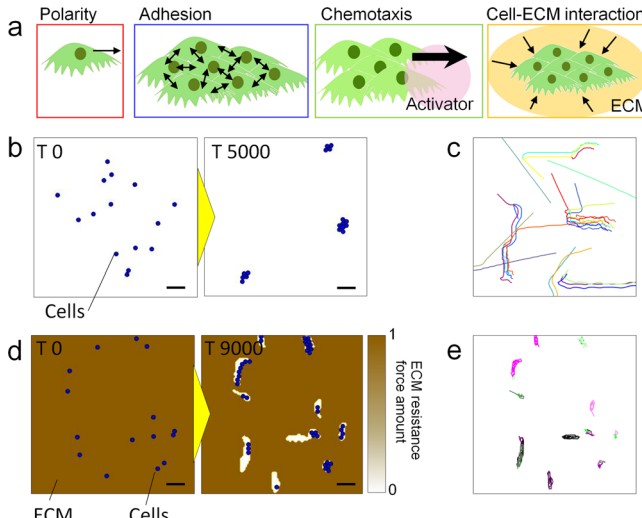

**Fig. 5 Computational simulation of cellular behaviour. a** Schematic representation of the physical meaning included in the mathematical model —directional polarity, cell–cell adhesion, chemotaxis, and cell–ECM interaction. **b** Simulation result of cellular behaviour without resistance force from ECM. **c** Particle tracking result of cells in Fig. 5b. **d** Simulation result of cellular behaviour with resistance force from ECM. The amount of ECM resistance force is continuously updated corresponding to cellular movement. **e**, Particle tracking result of cells in Fig. 5d. The complete time-lapse movies for parts **b** and **d** are shown in Supplementary Movies 10 and 11. Scale bar: 200 μm (**b**, **d**).

In spite of the lack of FN, the result that collective cells progressively invaded into the degraded ECM area, in contrast to when cells could not migrate outside of the circular shape at the beginning without artificial degradation, indicates that the effect of lack of FN site contributes less to hinder cell migration compared with the resistance force from ECM. In other words, both the surrounding ECM and lack of FN site prevented cells from migrating to new sites, but cells can easily cross the FN boundary when pushed out by the many surrounding cells as shown in Fig. 1f, whereas cells cannot overcome the resistance force from ECM until it is weakened enough. Again, a mathematical simulation was conducted based on the same model as that in Fig. 5d, but cells were placed in a circular area to correspond with the in vitro experiments. As a control, Fig. 6e and Supplemental Movie 14 shows the simulation result of cells covered by intact matrigel. Cells initially meandered around to form clusters and migrated rotationally within the initial circular shape, but eventually migrated outward from the circular shape. This behaviour agrees well with the in vitro results shown in Fig. 2e and Supplemental Movie 5. On the other hand, the simulation with artificial degradation of ECM, modelled by removing a section of ECM on the right side of the initial cell location, showed the cells progressively migrating towards the degraded area (Fig. 6f and Supplemental Movie 15). Cells on the right-hand side of the circular area moved slightly towards the area without ECM from the beginning, and with an increasing number of cells, they progressively migrated to fill the ECM-free space. Cells barely migrated to explore new sites, rather preferring to stay in lower resistance areas. This simulation result also showed good agreement with the experimental results in Fig. 6a and Supplementary Movie 13. Thus, we showed that active remodelling of the ECM can control the direction of cellular migration.

**Directing branch pattern formation by controlling local ECM stiffness**. The stiffness of ECM is one of the major factors that determine the resistance force, and is relatively easy to control by changing the ECM concentration or degree of crosslinking. Genipin, a cross-linker for proteins such as collagen and gelatine, can be used to modify the stiffness or viscosity of matrigel. The physical changes of matrigel depending on genipin concentration have been well described in the work of Imai et al.[57]. We employed this cross-linker to analyse the effect of ECM stiffness on cellular migration, as well as the 3D-branching pattern formation of lung epithelial cells. First, NHBE cells were cultured in 2D covered by matrigel with different concentrations of genipin for 48 hours, and time-lapse images were taken every 5 mins (Fig. 7a). The images were then converted into binary images so that the cell area could be identified in black. Then, all images were summed to obtain the total area in which cells had moved in. Comparing the rate of expansion of the cell movement area, there were significant differences between the cases with and without genipin (Fig. 7b and data are available in Supplementary Data 2), and higher concentration of genipin in matrigel-restricted cell movement.

This restriction of cell movement is likely to influence 3D morphology as well, and thus we also investigated the morphological changes in the 3D branching of bronchial cells. An NHBE clot with a high density of cells was co-cultured with human umbilical vein endothelial cells (HUVECs) dispersed in matrigel to develop branching formation in a 3D space, as has been described previously[46], with genipin mixed into the matrigel to alter its stiffness. The branches that typically develop in control conditions were largely restricted by the addition of genipin, and the size of the clot also became much smaller than the initial clot size (Fig. 7c). The restriction of branching was considered to be associated with an increase in resistance force by the increase in ECM stiffness; cells did not have enough force to carve the ECM when the resistance force from ECM was too high, thus hindering the generation of branches.

Next, in order to localise regions with and without genipin within the same gel, a scratch was made on a dish to enhance the surface tension to sustain the hydrogel on the edge of the scratch, and uncured matrigel with and without genipin were placed on either sides of the scratch. After curing the matrigel, genipin concentration and thus stiffness of the hydrogel could be localised, but because the hydrogel had been seamlessly combined, cells are able to cross the boundary between hydrogels of different stiffness (Fig. 7d). By employing this hydrogel localisation technique, an NHBE clot was cultured at the boundary of matrigel with and without genipin, with HUVECs evenly dispersed in the matrigel surrounding the clot. Initially, cells aggregated on both sides, but after 10 hours, branches started to form in the area without genipin, whereas cells in the area with genipin migrated towards the area without genipin. After 30 hours, many branches had developed in the area without genipin but no branches had formed in the area with genipin. Cells in the area with genipin also migrated towards the area without genipin, and the clot size in the genipin area shrank much more than that in the area without genipin (Fig. 7e and Supplementary Movie 16). We also confirmed that NHBE cells produced FN and left their footprint on the ECM during branching morphogenesis (Fig. 7f), as they did in 2D migration (Fig. 3e). The results have shown that cells in the higher stiffness area were not able to carve the ECM, resulting in restriction of branching morphogenesis; instead, cells migrated towards the lower resistance force area where other cells had been present before and had marked the area with FN and loosened the ECM.

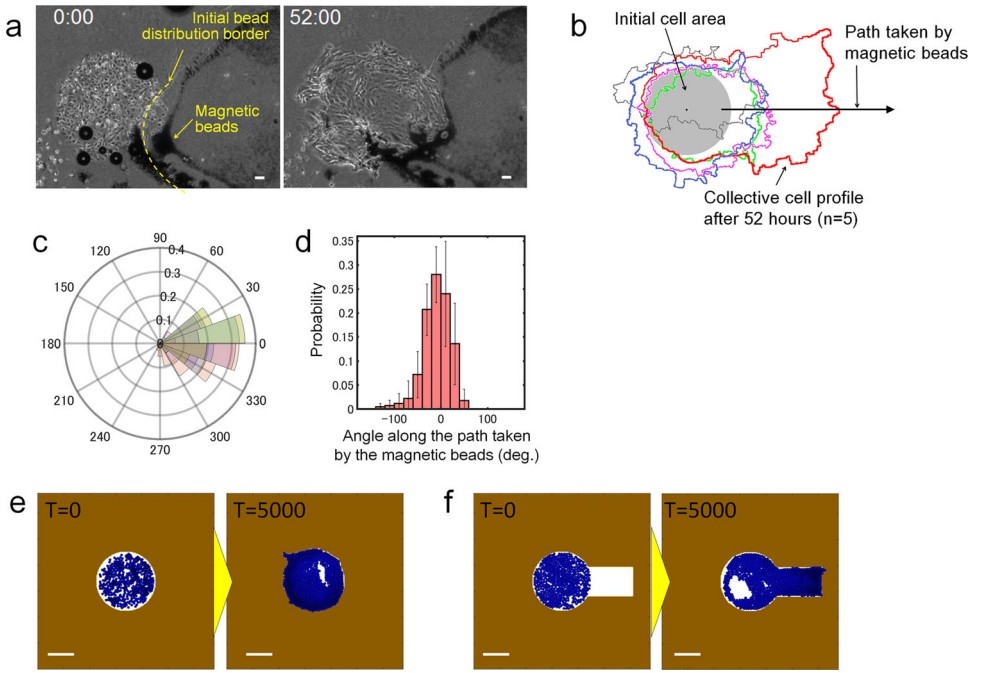

**Fig. 6 Effects of artificial modification of ECM resistance force on the directionality of collective cell migration. a** Time-lapse images of collective cell migration when the ECM was artificially modified by magnetic bead manipulation. Repetitive manipulation of magnetic beads by a permanent magnet partially weakened the resistance force from the ECM at the specific location. **b** Collective cell boundary after 52 hours. Five culture experiments were conducted ($n = 5$) and each colour pattern represents the result of one collective cell profile. Cells were initially patterned in the shape of a circle, but many of the cells migrated towards the area weakened by the magnetic beads. **c** Polar histogram of the probability of angle of cell location from the centre of the initial circular shape after 52 hours. Five culture experiments were conducted ($n = 5$). **d** Probability of the movement of cells out of the initial circular area as a function of the angle along the axis of the path of the magnetic beads. Approximately 75% of the cells migrated towards the area degraded by the magnetic beads. Five culture experiments were conducted ($n = 5$), and on average over 30,000 data points per experiment were obtained to calculate the probability. The data shown are the means ± SDs ($n = 5$). **e** Computational simulation result without pre-made degradation area in the ECM surrounding the collective cells. **f** Computational simulation result with pre-degraded area in the ECM surrounding the collective cells. The complete time-lapse movies for Fig. 5a, e and f are shown in Supplementary Movies 12–14. Scale bar: 100 μm (**a**), 500 μm (**e**, **f**).

## Discussion

Collective epithelial cell migration is an essential organisational process in the formation of complex tissues such as the lung and the kidney. In this work, we analysed how cells alter their microenvironment, and how this ECM modification influences the directionality of cellular migration. A variety of engineering technologies based on microfabrication and optical tweezers were employed to investigate the effect of ECM modification in a quantitative manner. Spatio control of the microenvironment surrounding cells provide a clear boundary between cells and ECM. Then, dynamic analysis of cellular behaviour was conducted based on time-lapse imaging and mathematical simulations. The results with spatio control of cells show that cells deposited FN and constructed adhesion sites in the areas in which they had previously existed. The time-lapse imaging results with dual patterning of cells and FN area revealed that cells preferentially remained within the constructed site instead of exploring new sites; however, once the constructed adhesion site becomes too crowded, the cells are able to migrate out of the site. In addition, there is a resistance force from the ECM at the boundary between the cells and the ECM that keeps the cells in their original site, and it is thought that this resistance force is dynamically modulated to regulate the directionality of collective cell migration. For example, the physical movements of cells can alter the resistance force from ECM so that cells can expand their territory by themselves. Our experiments using optical tweezers and microbeads to emulate the physical movement of cells, as well as simulation results of mathematical models which take into account cell–ECM interactions in addition to cell polarity,

cell–cell interactions, and chemotaxis, support our hypothesis of the dynamic remodelling of ECM resistance force by cells and the contribution of this dynamic remodelling of ECM resistance to the direction of cellular migration and pattern formation. Furthermore, we showed in vitro that artificial modifications of the ECM using magnetic beads and genipin cross-linker can direct collective cell migration, with cells tending to migrate towards areas with lower resistance force rather than explore areas with higher resistance force. In fact, branch patterns from bronchial epithelial cells were generated only in the softer ECM area when we controlled the distribution of ECM stiffness. Multicellular organisation is orchestrated by the diverse composition of cell and ECM, and optimal ECM stiffness depends not only on the cell type and organ function, but also on the developmental stage of the tissue. Therefore, a soft and low resistance ECM is not always suitable for cells to migrate. There is a potential to direct tissue pattern formation by controlling the distribution of ECM stiffness. In vivo, the stiffness of ECM surrounding organs is not uniform; for example, ECM surrounding the trachea has a higher stiffness than that surrounding the bronchus owing to the existence of cartilage and thickness of ECM. Yet, most of the prevailing in vitro experiments have been conducted under conditions of uniform ECM properties. For example, the immaturity of the structure and functionality of an organoid model integrating multiple organs to link the liver, biliary, and pancreas, which was reconstituted in vitro in a uniform matrigel environment could be attributed to the differences in ECM composition and stiffness between the different organs in vivo. On the other hand, the technology demonstrated in this paper with a simple

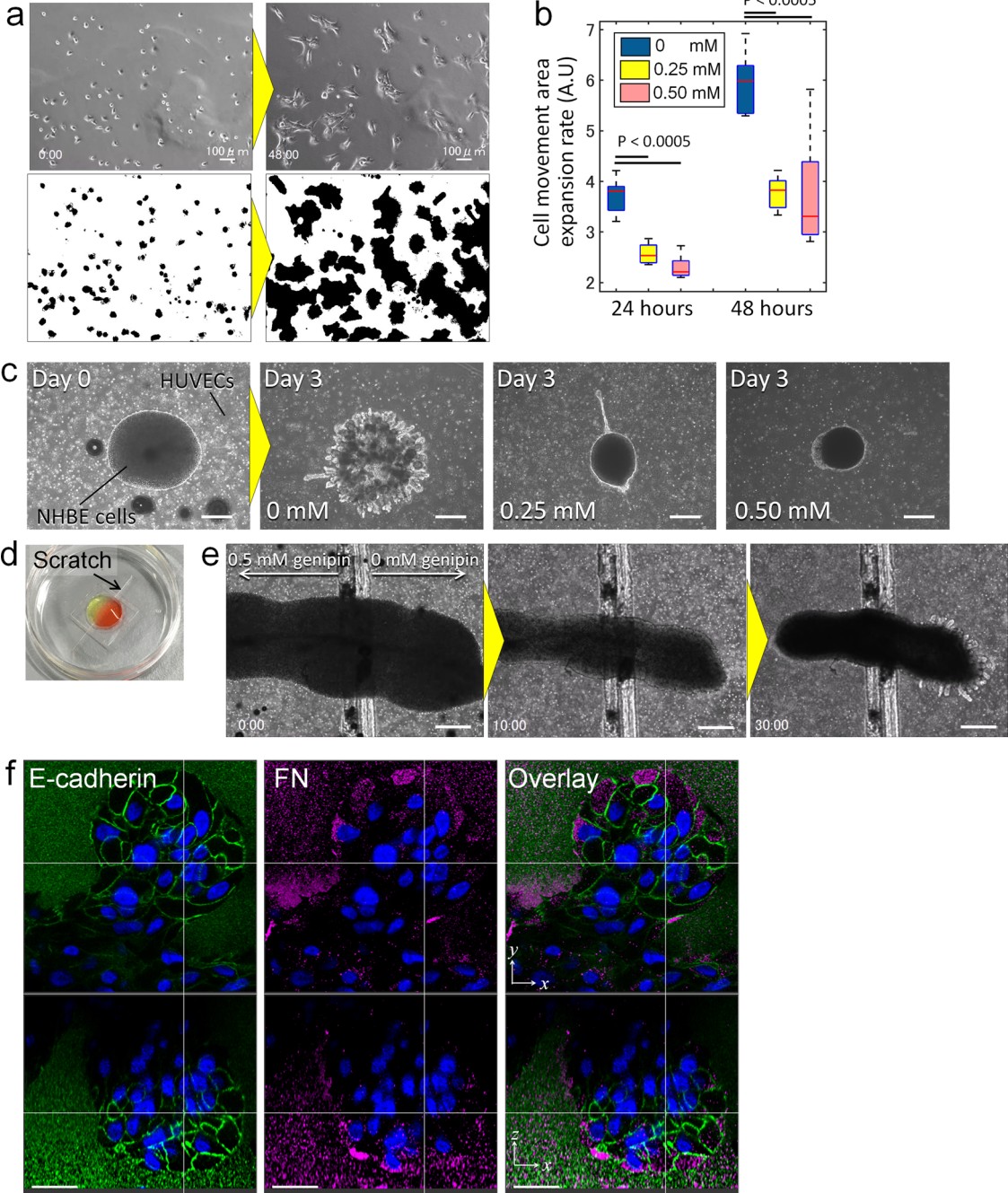

**Fig. 7 Effects of ECM stiffness on cellular behaviour and 3D morphogenesis. a** Time-lapse images of NHBE cells on a dish covered by matrigel with genipin to enhance the stiffness of matrigel. The images were converted to binary to calculate the area of cellular movement. **b** Quantitative comparison of cell movement area with different genipin concentrations after 24 h and 48 h. The data shown are the means ± SDs ($n = 9$). **c** Phase contrast images of 3D cultured NHBE cells. HUVECs were dispersed in matrigel while a high density of NHBE cells was placed in the centre of the matrigel. Genipin was mixed with matrigel at a concentration of 0, 0.25, and 0.5 mM to analyse the effects of ECM stiffness on the branching morphogenesis of NHBE. **d** Demonstration of ECM localisation by a scratch made on the culture dish. Yellow- and red-coloured matrigel were localised on either side of the scratch. **e** Time-lapse images of 3D culture of NHBE cells. The left part of the image shows matrigel with 0.5 mM genipin while the right part of the image shows matrigel that does not contain genipin. **f** Section imaging of a developed branch in matrigel showing E-cadherin, FN, and nucleus staining. The complete time-lapse movie for part a is shown in Supplementary Movie 16. Scale bar: 100 μm (**a**), 500 μm (**c**, **e**), 30 μm (**f**).

scratch method on a dish to localise ECM with two different stiffness, indicate that it is important to develop technologies to control culture conditions, including ECM localisations, to reconstitute more sophisticated tissue formations in the future.

In this paper, we analysed only the physical interaction between cell and ECM by employing optical tweezers to discuss the universal phenomenon, but the secretion of matrix

metalloproteinases (MMPs) by specific types of cells also contributes to the degradation of ECM and the weakening of the resistance force. The functions of the MMP family and their contributions to cellular migration have been well-discussed in many previous studies[58–60]. In 3D-branching morphogenesis of NHBE cells, we have observed that a leader cell carved the surrounding ECM to make a tunnel, just like NHBE cells did in 2D

as shown in Fig. 3e, prior to the extension of branches (Supplementary Fig 5. and Supplementary Movie 17). Dye-quenched collagen IV, which shows degraded collagen, was employed to visualise the tunnel, and higher signal of fluorescence was observed at the peripheral tunnel, which suggests the presence of MMP. Thus, the identification of the specific MMPs for different cell types and ECM is required to consider both physical and chemical interactions for further elucidation of the mechanisms of dynamic remodelling during cellular migration. In the present model, the cell–ECM boundary was simplified to a gradient of the ECM field to quantify the resistance force. However, at the cell level, other mechanical remodelling events also occur in the ECM, for example, fibre re-alignment owing to cell traction force and the orientation of surrounding cells[61–63], and these events also have an effect on cellular migration[64], although they likely occur prior to the degradation of the resistance force. Interface instability, particularly where boundary conditions change steeply between the cell–ECM and ECM-tunnel region, is also likely involved in the current system, and is thought to contribute morphogen diffusion to cause chemotaxis. In future work, these events should be taken into account with experimental validation prior to the more comprehensive analysis of cell–ECM interaction.

Collective cell migration driven by mechanical forces[17,18] generated by cell–cell interactions and morphogen gradients[10,15,22] have been considered to be the systems that govern pattern formation; however, our results suggest that local remodelling of the microenvironment surrounding cells also considerably contributes to the directionality of the collective migration. By incorporating ECM-remodelling events into existing theoretical models, such as the vertex models[65–67] and reaction-diffusion models[68–70], the accuracy of mathematical simulations can be improved significantly. As the interplay between chemotaxis and mechanotaxis has an important role in self-organised pattern formation, our findings on the effects of ECM remodelling on cellular migration provide additional insights into the role of the ECM in complex pattern formations, such as in the branching of lung airway and kidney tubules.

## Methods

**Cell culture**. NHBE cells and HUVEC were obtained from LONZA (Walkersville, MD, USA) and cultured using the BEGM Bullet Kit (CC-3170; LONZA) and EGM-2 Bullet Kit (CC-3162; LONZA), respectively, supplemented with 50 IU/ml penicillin and 50 μg/ml streptomycin (Thermo Fisher Scientific Inc., MA, USA). For the consistency of cell viability, the use of both cell types was limited to five passages.

**Dual patterning of cell aggregates and FN**. Photolithography was employed to produce a shadow mask membrane in order to pattern cells on a dish. Before the fabrication process, a silicon substrate was soaked with a mixture of hydrogen peroxide and sulphuric acid (1:2) for 10 min to clean the substrate, followed by rinsing with deionised water for 5 min twice. Then, the substrate was dried in the oven at 140 °C for 20 min. A TDFS300 (DJ Microlaminates, MA, USA) negative thick dry photoresist was laminated onto the silicon substrate at a thickness of 300 μm. A photomask with the desired geometric pattern was then placed on the photoresist and exposed to UV, followed by a developing process using 2-methoxy-1-methylethyl acetate (Wako Pure Chemical Industries, Ltd., Osaka, Japan). Once the photomask pattern has been transferred to the photoresist, the silicon substrate was used as a mould for producing a PDMS membrane with the pattern. PDMS was poured onto the mould at a thickness lower than that of the mould and incubated in the oven at 90 °C for 20 min. After the PDMS membrane had been cured, it was detached from the silicon mould, and 70% ethanol was sprayed on the membrane and a culture dish. The membrane was immersed in 50 μg/ml bovine FN (F4759: Millipore Sigma Corporation, MO, USA), then placed on a dish before the FN dried out, and the dish was incubated at 37 °C for 20 min. Subsequently, a cell suspension medium was applied over the membrane in the unmasked area, and the dish was incubated at 37 °C for 4 h. The PDMS membrane was then removed for further experiments.

**2D migration assay covered by matrigel**. The same procedure as above was conducted to produce a PDMS membrane. The membrane was set on the dish before the ethanol dried out. Adhesion of the PDMS membrane to the dish is enhanced

once the ethanol has been dried out in an 80 °C oven for 10 min, so that leakage can be prevented in the following processes. Cell suspension medium was applied over the membrane in the unmasked area, and the dish was incubated at 37 °C for 4 h. The PDMS membrane was then removed, and growth factor-reduced matrigel (356231: Corning Incorporated, NY, USA) was applied over the cells. After gelation, a one-to-one mixture of BEGM and EGM-2 (B + E medium) was added to the dish.

**3D culture experiments of NHBE cells to induce branching morphogenesis**. NHBE cells and HUVECs were co-cultured to develop a bronchial tree[46]. HUVECs were dispersed in growth factor-reduced matrigel (#356230; Corning, NY) with different concentrations of genipin (G4796; Millipore Sigma; MA, USA), with a cell density of $3.0 \times 10^3$ cells/μL of matrigel, then gently mixed with a pipette. Two hundred microliter of mixed matrigel was then injected on a $\phi$ 12-mm cover glass placed on a 12-well plate to sustain a sufficient matrigel height. Next, condensed NHBE ($5.0 \times 10^4$ cells/μL) taken from the pellet of cells obtained after centrifugation was injected into the centre of the HUVEC-matrigel gel, and the sample was incubated at 37 °C for 25 min to facilitate gelation. Two millilitres of B + E medium was added to each well, and the medium was changed every day.

**Movability and resistance force measurement in the ECM**. An optical tweezer system was used to measure the movability and resistance force in the ECM. The wavelength of the laser was 1064 nm and the output power through a ×100 objective lens measured by a laser power metre (ORION-PD; Ophir Optronics Solution Ltd, Jerusalem, Israel) was 600 mW. Using this system, 10 μm polystyrene beads were manipulated in matrigel and collagen I. Initially, the crosslinking of matrigel was disrupted by the beads. Thus, the microbeads were manipulated very slightly (~10 μm) around the beads by using this setup, although bead control was lost for manipulations over longer lengths. To analyse dynamic modulation of the movability of micro-objects in the ECM by repetitive movement, the laser was scanned back-and-forth for 100 cycles, and the length between the point at which the beads were trapped to that at which they were released was measured after every 20 cycles of laser scanning. The detailed method for the calibration of the optical tweezers to calculate the resistance force in the ECM is described in the Supplementary Information.

**Cell tracking**. Live cell images were acquired with a BZ-X700 microscope (Keyence; Osaka, Japan) equipped with a stage-top incubator (Tokai Hit Co. Ltd.; Shizuoka, Japan) every 5 min. Subsequently, cell movement was tracked manually, and cellular behaviour visualised using the MATLAB software (MathWorks, Inc., Natick, MA). A hundred images of every 10 min from a time-lapse video were loaded into the figure command sequentially, and the cells were marked to record their positions on the x–y axis. The positional data from the tracking were converted to a vector format and visualised.

**3D Immunostaining imaging**. Immunostaining and confocal imaging were conducted to visualise laminin distribution. For fixation, 4% paraformaldehyde (Wako Pure Chemical Industries, Ltd.; Osaka, Japan) was applied to the matrigel-containing cells at room temperature for 20 min, followed by two rinses with phosphate-buffered saline (PBS), 10 min for each rinse. The sample was permeabilized with PBS containing 0.5% Triton X-100 for 10 min at 4 °C, followed by three washes with PBS containing 100 mM glycine,10 min for each wash. The samples were then incubated with 10% goat serum in IF-buffer (0.2% Triton X-100, 0.1% BSA, and 0.05% Tween-20 in PBS) for 60 min at room temperature as a primary blocking step. As a secondary blocking step, the samples were incubated with 1% goat anti-mouse immunoglobulin G and 10% goat serum in IF-buffer for 40 min at room temperature. To stain laminin, the sample was incubated with a rabbit polyclonal anti-laminin antibody (1:100, L9393 Millipore Sigma; MA, USA) overnight at 4 °C, followed by three rinses with IF-buffer, 20 min for each rinse. A secondary antibody-labelled with Alexa fluor 594 (1:100, Thermo Fisher Scientific Inc.; Waltham, MA) was then applied. For nuclear staining, the samples were incubated with 300 nM DAPI in PBS for 20 min at room temperature, followed by three rinses with PBS, 20 min for each rinse. Then, three-dimensional images were taken with SP8 lightning confocal microscope system (Leica Microsystems; Wetzlar, Germany).

**Mathematical model of cellular behaviour**. A particle model has been developed to simulate cellular behaviour in the ECM. The model follows the same principles as that of Akiyama et al.[55], but with the additional parameter of a resistance force from the surrounding environment. A detailed explanation of the model is described in Supplementary Information.

**Artificial degradation of the ECM by magnetic beads**. Magnetic spherical microbeads with a 5 μm diameter (MPS5UM: Magsphere Inc., CA, USA), which contain iron oxide crystals in a polystyrene matrix, were used to mimic collective cell migration and to artificially degrade the ECM. Matrigel was diluted by half with culture medium to enable the manipulation of magnetic beads using a permanent magnet under a glass-bottom dish. 0.625% w/v magnetic microbeads were gently mixed with the matrigel, then applied over the patterned NHBE cells in a glass-

bottom dish. Before gelation, a neodymium permanent magnet that was $10 \times 10 \times 15$ mm in size (NeoMag Co., Chiba, Japan) and that had a surface magnetic flux density of 511 mT was placed under the dish to collect and align the magnetic beads close to the cells. Then, the cells were incubated at 37 °C for 25 min for gelation to occur. Subsequently, the magnetic beads were manipulated back-and-forth by the permanent magnet repetitively to mimic collective cell migration and to degrade the ECM. After most of the beads have been moved away from the cell accumulation site, time-lapse imaging was started to observe the directionality of collective cell migration of the NHBE cells.

**Quantification of the directionality of collective cell migration using polar histograms.** The directionality of collective cell migration towards the area of ECM that has been degraded by magnetic beads was quantified. First, phase-contrast time-lapse images (Supplementary Fig. 4a) were converted into binary images using the "Find Edges" function in ImageJ to properly identify the boundary of the cell aggregates. Subsequently, background noise was subtracted using the "open" and "close" function repetitively in the binary option. Once a clear binary image was obtained (Supplementary Fig. 4b), the original circular area, where the cell aggregates were placed at the beginning, was subtracted from the binary image (Supplementary Fig. 4c). The images were then analysed in MATLAB to calculate all the angles ($\theta_n$) of the black grids from the original centre circle against the axis of the path of the magnetic beads (Supplementary Fig. 4d). A polar histogram of probability values was generated to show the directionality of collective cell migration.

**Statistics and reproducibility.** All sample sizes are larger than five and specified in figure legends. Statistical significance, determined using Welch's $t$ test or student $t$ test, was calculated in Matlab or Microsoft Excel.

**Reporting summary.** Further information on research design is available in the Nature Research Reporting Summary linked to this article.

## Data availability
All data generated or analysed during this study are included in this published article (and in Supplementary Data 1 and 2).

## Code availability
Computer code used to simulate the cellular behaviour was developed in Matlab and available from the corresponding author on reasonable request.

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

## Acknowledgements

This work was financially supported by JSPS KAKENHI (18H04765, 18H01413).

## Author contributions

M.H. and I.K. carried out the in vitro experiments and microfabrication. H.M. conducted the non-contact manipulation of microbeads by optical tweezers. H.M. and F.A. analysed the obtained data to calculate the resistance force. M.A. and M.H. developed a mathematical model and conducted simulation. M.H conceived and designed the experiments and wrote the manuscript. All the authors have read and approved the final manuscript.

## Competing interests

The authors declare no competing interests.
