## [Peer Review File · Communications Biology]

Reviewers' comments:

Reviewer #1 (Remarks to the Author):

In the manuscript at hand, Hagiwara and colleagues analyze the spatio-temporal remodeling of ECM surrounding collective epithelial systems. In particular, how this alters the resistance force exerted on cells. They authors show how the cells construct and break the ECM during their migration and how the ECM can guide cellular movement.

While the content and results of the work is interesting, I believe a MAJOR revision is necessary. The detailed comments are listed below.

Major Points

1. Why do the cells aggregate to the lower right corner during the experiment of confinement with the triangular PDMS stencil? This peculiar symmetry breaking is not explained by the topology of the system and rather points to some coating artifact.
2. In the experiments with the Matrigel coating, it would be interesting to see the cell organization inside the Matrigel layer. The initial area covered by the cells decreases, and z-stacks of the monolayer would shade light on how the cells move inside the Matrigel. Simple horizontal projection on the surface are not sufficient to provide the entire picture.
3. In the experiments with the Matrigel, a hypothesis is being made regarding the resistance forces exerted by the ECM. The hypothesis is then validated by tracking the movement of the cells, as shown in figure 3. The tracking doesn't really display the resistance of the ECM. Please perform an experiment of traction force microscopy (TFM - several simple approaches exist) and compare the traction that the cells apply on the surface in the two cases (with and without Matrigel).
4. In the results regarding the quantitative measurement of the resistance force, why is there very little to no difference in the x direction?
5. The 3D movement of the cells in the ECM is neglected. While the carving of the ECM is reported, it is only analyzed in 2D.
6. The experiment showed in figure 6 is redundant. It was already shown that cells expand more in absence of the Matrigel. I think this part of the results can be removed since the experiments with the different Matrigel reporting various concentrations of genipin is sufficient to make the point.
7. In the conclusions you state that in vivo the stiffness of the ECM surrounding organs is not uniform. Could you please add a part in the results and in the figures where you measure and report the stiffness of real tissues? Once knowing the different regions of stiffness or by using two tissues that have different stiffness, it would be possible to repeat the experiments of cell migration by decellularizing the tissues and by seeding the cells in the remaining ECM? This way it would be possible to prove your findings from the in vitro experiments.

Reviewer #2 (Remarks to the Author):

In this work, the authors systematically investigate how the spatial-temporally remodelled extracellular matrix (ECM) alters the resistance force exerted on cells during collective migration via a synergy of combined experimental and modeling efforts. In particular, in vitro experiments were carried out to demonstrate the simultaneous construction and breakage of ECM during cellular movement, and to show that this modification of the surrounding environment can guide cellular movement. By artificially remodelling the microenvironment, it was showed that the

directionality of collective cell migration, as well as the 3D branch pattern formation of lung epithelial cells can be manipulated.

The paper is overall very well written and easy to follow by both biologists and physicists. The study is comprehensive and the results are overall sound. I am happy to recommend its publication, once the authors constructively address the following concerns and comments:

If I understand correctly, the key observation made was that a cell colony/cluster with engineered shape/symmetry initially felt a high level of resistance force on the colony-ECM interface. This resistance force was subsequently weakened, as cellular movement carved ECM, creating micro-channels leading to particularly correlated migrations. Similar behaviors were recently observed in collective invasion of engineered colonies of MDA-MB-231 cells with a varieties of geometrical shapes and symmetry (including circle, triangle, star and half-circle) in 3D ECM (see, e.g., Kim, et al *Biophysical Journal* 118, 1177 (2020)). There the forces were not directly measured but inferred from the deformation field of the ECM surrounding the cell colony. Initial compression deformation was observed, which was relaxed as cell invasion started, and eventually become tensile deformation. The relaxation rate depends on the direction along different symmetry axis of the colonies, along which the invasion was also different. The compression-tension transition was shown due to strong pulling of migrating cells.

It seems that in the computational model, the influence of the ECM on individual cells is treated as a resistance force field that depends on the dynamic states of the cells (in a relatively ad hoc manner) so that it can properly pick up the effects of ECM degradation due to cell movement. Although it is reasonable to assume approximately smooth and continuous force field on large scales (e.g., on the level of the entire colony), it is not quite clear whether on the cell-level scale these quantities can be treated as smooth field with well defined gradients (as used in the definition of resistance force). On the cell level, recent studies showed other important cell-ECM remodeling and feedback mechanisms that can regulate collective migration (Fan et al., *Angewandte Chemie*, doi/10.1002/anie.202016084). Computational models that more explicitly treats cell-ECM mechanical interactions were also reported (*Physical Biology* 13, 066001 (2016); *Physical Review E* 100, 043303 (2019)), which suggest cells feel very fluctuating and heterogeneous forces transmitted via ECM fibers.

The authors showed very interesting experimental and simulation results for collective migration in artificially remodeled ECM. Previous investigations taking into account ECM degradation and mechanical interactions between cell colony and ECM showed that ECM degradation by actively migrating cells not only lead to micro-channels for subsequent follower cells, but also result in certain type of interface instability in the low ECM density regions (*AIP Advances* 2, 011003 (2012)). Would such instability also occur in the current system?

Reviewer #3 (Remarks to the Author):

This paper is focused on role of cell-ECM interactions in collective cell migration. The authors demonstrate that weakening ECM allows preferential movement paths for cells. The creative approach employs microfabrication and non-contact manipulation methods to dynamically and locally modify the matrix. The results are compelling but some issues should be addressed. Explain under what conditions do epithelial cells collectively migrate through ECM? Are there cases where these cells are not effectively migrating on 2D surfaces of organs to cover them, or where these cells move into 3D solid tissues without the way being cleared by supporting cells that secrete MMPs and remodel the matrix. Also, I understand that this paper is focused on bronchial epithelium, but this specific type of collective migration and invasion seems highly related to vasculogenesis and angiogenesis which are never mentioned. The vast research in this area is not cited. If it is not related, maybe explain why. Many questions arise about the use of modeling in this manuscript. The model seems designed to support the hypothesis rather than test or inform it ("developed to back up the above hypothesis"). Please clarify how cell polarity is not directly coupled to chemotaxis (2 of the 4 facets

of the model). And what about cell-ECM adhesion in terms of anchoring/traction? The presence and density of specific ECM ligands and their respective cell receptors (integrins) on the cell modulates cell-ECM interactions also, in addition to the ECM resistance or barrier function described here.

On a related topic, later in the discussion ECM stiffness is considered but only in the sense that it is suggested that less stiff matrix would offer less resistance to migration. However, matrices have a diverse range of stiffness that is optimized for function. In fact, matrix that is too soft would not support migration at all because the traction forces could not be generated at adhesion sites. In the figure 6 experiments and modeling, why was a large area cleared instead of making single bead tracks which would more realistically mimic the action of cells on the matrix? Also discuss how did the lack of fibronectin deposition (as would be done by cell action) in the artificially remodeled space affect cells migrating into the area? Finally, there is heavy, almost excessive, self-citation.

Physical interaction of cells with ECM weakens the resistance force and regulates the directionality of collective cell migration and pattern formation

by Masaya Hagiwara, Hisataka Maruyama, Masakazu Akiyama, Isabel Koh, Fumihito Arai

Manuscript No. COMMSBIO-21-0550

On behalf of all the authors, I would like to thank the reviewers for taking the time to carefully review our paper, and for their insightful comments. The submitted manuscript (COMMSBIO-21-0550) has been revised based on the reviewers' comments.

The revised parts are highlighted in the manuscript attached at the end of this file, and the point-by-point responses are as follows:

Responses to Reviewer #1

1-1

In the manuscript at hand, Hagiwara and colleagues analyze the spatio-temporal remodeling of ECM surrounding collective epithelial systems. In particular, how this alters the resistance force exerted on cells. They authors show how the cells construct and break the ECM during their migration and how the ECM can guide cellular movement.

While the content and results of the work is interesting, I believe a MAJOR revision is necessary. The detailed comments are listed below.

Why do the cells aggregate to the lower right corner during the experiment of confinement with the triangular PDMS stencil? This peculiar symmetry breaking is not explained by the topology of the system and rather points to some coating artifact.

Response 1-1;

We appreciate the reviewer's comprehensive comments.

When NHBE cells were seeded in a triangular shaped-pattern on a dish and covered by Matrigel, cells initially migrated toward to the tip area, leaving the center of the triangle empty. This is due to the morphogen distribution of short-range activator and long-range inhibitor, which are secreted by lung bronchial epithelial cells to communicate with other cells to form a specific pattern in 2D and 3D. The reason cells migrate towards the tip area has been explained, as follows, in our previous paper. "The effect of the inhibitor is widespread due to the high diffusion rate. Therefore, the inhibitor concentration is higher at the center of the collective cells because the number of surrounding cells is higher while the concentration is lower at the tips of the collective cell geometries because of the presence of a much lesser number of surrounding cells. On the other hand, the effect of the activator is spatially limited due to its low diffusion rate and its concentration does not depend on the number of surrounding cells. Then, cells tend to move to either the tips or edges of the collective cell geometries to avoid areas containing a high concentration of inhibitor, and directional migration of the

collective cells was generated at the tips first”

As the high peak of activator concentration at the tip starts to diminish, cells begin to meander inside the triangle shape. This is the timing shown in Fig 1b, which is after 24 hours of culture. In order to focus on the part of fibronectin production, we had skipped the explanation of NHBE cellular behaviour, but as Reviewer 1 pointed out, it is also an important dynamic of NHBE cells. Thus, we have added a brief explanation of the above, with the citation of our previous work, in page 3.

1-2

In the experiments with the Matrigel coating, it would be interesting to see the cell organization inside the Matrigel layer. The initial area covered by the cells decreases, and z-stacks of the monolayer would shade light on how the cells move inside the Matrigel. Simple horizontal projection on the surface are not sufficient to provide the entire picture.

Response 1-2;

We appreciate the reviewer’s suggestions.

We have obtained z-stack images to visualize the cellular organization after NHBE cell seeding, at 24 hours of culture with and without Matrigel cover, and at 48 hours of culture with Matrigel cover. These were added to Figure 2g, with the whole 3D voxel images shown in supplemental video 8. The 3D image results surely provide more information on how cells reorganize when they are aggregated and released. The following result and description have been added in pages 5 and 6.

“In order to analyse cellular organization with and without Matrigel cover, z-stack images were taken after NHBE cells were seeded in a circular pattern, at 24 hours with and without Matrigel cover, and at 48 hours with Matrigel cover (Fig. 2g and Supplementary video 8). Initially, cells still retained their individual, rounded shape just after seeding, but are in contact with each other on the glass bottom surface of the dish. After 24 hours of culture without Matrigel cover, as cells expand outward as shown in Fig. 2d, they flatten into a thin monolayer. On the other hand, after 24 hours of culture with Matrigel cover, as cells aggregated inside the original circular pattern as shown in Fig. 2e, the cells were observed to accumulate into 2-3 layers. This modification of cellular organization is thought to be due to cells being squashed by the surrounding Matrigel as well as being pushed by neighbouring cells. In fact, after 48 hours of culturing with Matrigel cover, as cells recovered to the original circular shape and started to migrate outside of the circular shape, cells transitioned back to a monolayer formation.”

Fig. 2g:

z-stack images to visualize the cellular organization after NHBE cell seeding, at 24 hours of culture with and without Matrigel cover, and at 48 hours of culture with Matrigel cover.

1-3

In the experiments with the Matrigel, a hypothesis is being made regarding the resistance forces exerted by the ECM. The hypothesis is then validated by tracking the movement of the cells, as shown in figure 3. The tracking doesn't really display the resistance of the ECM. Please perform an experiment of traction force microscopy (TFM - several simple approaches exist) and compare the traction that the cells apply on the surface in the two cases (with and without Matrigel).

Response 1-3

As Reviewer 1 pointed out, the tracking does not directly reflect the resistance force. The purpose of Figure 3 is to clarify the existence of a tunnel in the ECM which were made by the cells. Because of this tunnel, the resistance force from ECM will be weakened, and other cells can migrate along the same path. To avoid misunderstandings, the first sentence in the section in page 6 was modified.

TFM is surely prevailing method to measure the force applied on the substrate by cells, but because the cells are sandwiched between the glass bottom and Matrigel, traction force is generated on both sides. Under the current assay in this paper, TFM cannot directly measure the resistance force from the ECM. Instead, the quantification of the resistance force from ECM and its degradation was conducted using optical tweezers as shown in Fig. 4.

1-4

In the results regarding the quantitative measurement of the resistance force, why is there very little to no difference in the x direction?

Response 1-4

The microbeads actuated by optical tweezers were moved in only in the y-direction, as shown in Fig. 4b. Therefore, the majority of the force exerted on the ECM by microbeads is applied to push the ECM in the y-direction, while little shear stress and traction force were applied in the x-direction. The explanation above was added in page 7.

1-5

The 3D movement of the cells in the ECM is neglected. While the carving of the ECM is reported, it is only analyzed in 2D.

Response 1-5

We have added the new results of 3D branching morphogenesis of NHBE cells showing a leader cell carving the ECM prior to branching morphogenesis. Time-lapse imaging of NHBE cell movement in 3D culture was first carried out, and it showed that a leader cell separated from the follower cells and kept migrating in the ECM. In the Matrigel, fluorescent dye-quenched protein substrates (DQ-collagen IV) was added to visualize the path of the leader cell, and z-stack imaging was conducted after the

time-lapse experiment. The behaviour of the leader cell was the same as was already reported in our previous study in reference 46, but the z-stack images revealed that the leader cell created a tunnel in the Matrigel, just like the cell in 2D did. The results show that cells carved the ECM during migration in 3D as well. However, the fluorescence of DQ collagen is associated with the expression of MMP, which is not targeted in this paper. Therefore, we have added the results in the supplementary figure and the above explanation was added in the discussion section in page 12. For better understanding, a video from the time-lapse and z-stack images was added as Supplementary video 17.

Supplemental Fig. 5:

Time-lapse and z-stack imaging of 3D NHBE branching with DQ collagen in Matrigel. **a**, time-lapse imaging of NHBE branching in Matrigel supplemented with DQ collagen IV. Leader cells were emerged and branches were generated rapidly from NHBE cell clot. Then the leader cells were shot out from the branch while the cell path was remained in the Matrigel. **b**, xy spatial tiling image at the specific z height, taken by Z-stack imaging of the path of leader cells with DQ collagen. The tunnel was clearly observed at the path of the leader cells. The whole time-lapse and z-stack imaging is available in the supplementary video 17.

1-6

The experiment showed in figure 6 is redundant. It was already shown that cells expand more in absence of the Matrigel. I think this part of the results can be removed since the experiments with the different Matrigel reporting various concentrations of genipin is sufficient to make the point.

Response 1-6

The purpose of Figure 6 is to show the potential to control the directionality of collective cell migration and its pattern formation by controlling surrounding ECM conditions. One of the aims in this paper is to show the importance of designing the cellular environment and providing potential methodologies to control it in order to reconstitute tissue pattern formation, as is mentioned in the introduction. In particular, when it comes to the reconstruction of complex organoids in vitro, we believe that technologies to design and artificially remodel the ECM will be required. In that sense, the results from artificially degrading the ECM in a specific area by employing the external force from a magnet to direct collective cell migration is an important aspect in this paper.

1-7

In the conclusions you state that *in vivo* the stiffness of the ECM surrounding organs is not uniform. Could you please add a part in the results and in the figures where you measure and report the stiffness of real tissues? Once knowing the different regions of stiffness or by using two tissues that have different stiffness, it would be possible to repeat the experiments of cell migration by decellularizing the tissues and by seeding the cells in the remaining ECM? This way it would be possible to prove your findings from the *in vitro* experiments.

Response 1-7

We have not measured real tissue stiffness, but it is known that ECM surrounding the trachea has a higher stiffness due to the existence of cartilage rings and the thicker ECM compared to the subsequent bronchial branches and alveolar sacs. Similar to the response in 1-6, one of the aims in this paper is to show the importance of designing the cellular environment and providing potential methodologies to control it in order to recapitulate tissue pattern formation. Takebe et al. recently succeeded in developing a multiple organ integrated model to link liver-biliary–pancreas (*H. Koike et al., Nature 574. 2019*). Although the ECM composition and stiffness of each of those organs are different *in vivo*, the *in vitro* 3D culture was carried out in Matrigel with uniform concentration, which could be a reason why the developed structure is still immature. Our results show that ECM condition largely contributes to the behavior of collective cell migration and pattern formation, affirming that designing a proper ECM at the proper location is necessary to achieve sophisticated pattern formation. We had mentioned about trachea and bronchus stiffness in the discussion in order to emphasize this point, but have now expanded more on this explanation in page 12 to clarify our intention.

Responses to Reviewer #2

In this work, the authors systematically investigate how the spatial-temporally remodelled extracellular matrix (ECM) alters the resistance force exerted on cells during collective migration via a synergy of combined experimental and modeling efforts. In particular, in vitro experiments were carried out to demonstrate the simultaneous construction and breakage of ECM during cellular movement, and to show that this modification of the surrounding environment can guide cellular movement. By artificially remodelling the microenvironment, it was showed that the directionality of collective cell migration, as well as the 3D branch pattern formation of lung epithelial cells can be manipulated.

The paper is overall very well written and easy to follow by both biologists and physicists. The study is comprehensive and the results are overall sound. I am happy to recommend its publication, once the authors constructively address the following concerns and comments:

2-1

If I understand correctly, the key observation made was that a cell colony/cluster with engineered shape/symmetry initially felt a high level of resistance force on the colony-ECM interface. This resistance force was subsequently weakened, as cellular movement carved ECM, creating micro-channels leading to particularly correlated migrations. Similar behaviors were recently observed in collective invasion of engineered colonies of MDA-MB-231 cells with a varieties of geometrical shapes and symmetry (including circle, triangle, star and half-circle) in 3D ECM (see, e.g., Kim, et al Biophysical Journal 118, 1177 (2020)). There the forces were not directly measured but inferred from the deformation field of the ECM surrounding the cell colony. Initial compression deformation was observed, which was relaxed as cell invasion started, and eventually become tensile deformation. The relaxation rate depends on the direction along different symmetry axis of the colonies, along which the invasion was also different. The compression-tension transition was shown due to strong pulling of migrating cells.

It seems that in the computational model, the influence of the ECM on individual cells is treated as a resistance force field that depends on the dynamic states of the cells (in a relatively ad hoc manner) so that it can properly pick up the effects of ECM degradation due to cell movement. Although it is reasonable to assume approximately smooth and continuous force field on large scales (e.g., on the level of the entire colony), it is not quite clear whether on the cell-level scale these quantities can be treated as smooth field with well defined gradients (as used in the definition of resistance force). On the cell level, recent studies showed other important cell-ECM remodeling and feedback mechanisms that can regulate collective migration (Fan et al., *Angewandte Chemie*, doi/10.1002/anie.202016084). Computational models that more explicitly treats cell-ECM mechanical interactions were also reported (*Physical Biology* 13, 066001 (2016); *Physical Review E* 100, 043303 (2019)), which suggest cells feel very fluctuating and heterogeneous forces transmitted via ECM fibres.

Response 2-1

We appreciate the insightful comments of Reviewer 2.

As Reviewer 2 pointed out, the cell-ECM boundary in the model was simplified to a gradient to quantify the resistance force. At the cell-level scale, other mechanical and chemical remodeling events also occur in the ECM, such as fibre re-alignment due to cell traction force, orientation of surrounding cells, and MMP secretion. The purpose of the model is to illustrate the effects of the resistance force exposed on the cell-ECM boundary, and we believe the simplified boundary condition can satisfy our aim. However, the suggested papers surely describe the mechanical ECM remodeling during single cell migration, which was not taken into account in this paper. Also, the paper from Kim et al., illustrated how mechanical cues via fibre alignment affected cancer invasion. These ECM remodeling events probably also contributed to the degradation of resistance force described in this paper, and the inclusion of them in future mathematical models could lead to more comprehensive models.

We have added the above discussion, as well as suggested citations, in the discussion in page 13.

2-2

The authors showed very interesting experimental and simulation results for collective migration in artificially remodeled ECM. Previous investigations taking into account ECM degradation and mechanical interactions between cell colony and ECM showed that ECM degradation by actively migrating cells not only lead to micro-channels for subsequent follower cells, but also result in certain type of interface instability in the low ECM density regions (AIP Advances 2, 011003 (2012)). Would such instability also occur in the current system?

Response 2-2

We do believe that the effect of interface instability exists in the current system, especially since collective cell migration in NHBE cells is largely contributed to by chemical diffusion, and the morphogen secreted by cells are exposed on the interface between cell-ECM and ECM-degraded area. In this paper, we have not taken into account the interface instability due to the difficulties of experimental validation, but including the effect of this instability may lead to a more comprehensive analysis of the directionality of collective cell migration.

The above explanation was added in the discussion section in page 13.

Responses to Reviewer #3

This paper is focused on role of cell-ECM interactions in collective cell migration. The authors demonstrate that weakening ECM allows preferential movement paths for cells. The creative approach employs microfabrication and non-contact manipulation methods to dynamically and locally modify the matrix. The results are compelling but some issues should be addressed.

3-1

Explain under what conditions do epithelial cells collectively migrate through ECM? Are there cases where these cells are not effectively migrating on 2D surfaces of organs to cover them, or where these cells move into 3D solid tissues without the way being cleared by supporting cells that secrete MMPs and remodel the matrix. Also, I understand that this paper is focused on bronchial epithelium, but this specific type of collective migration and invasion seems highly related to vasculogenesis and angiogenesis which are never mentioned. The vast research in this area is not cited. If it is not related, maybe explain why.

Response 3-1

We appreciate Reviewer 3's comprehensive comments. As Reviewer 3 pointed out, this paper is focused on epithelial cells, in particular bronchial epithelium, which migrates within an epithelial monolayer during tube formation and undergoes various morphological events such as invagination and outgrowth into the ECM. Also, we agree that there are a lot of similarities between the collective cell migration in this system with that of endothelial vascularization and angiogenesis, while not many tip cells or leader cells are present during invasion into ECM in epithelial cell migration. In order to clarify the conditions of cell migrations and position of the paper, we have added the above explanation and several citation relating to vascularization research in page 2.

3-2

Many questions arise about the use of modeling in this manuscript. The model seems designed to support the hypothesis rather than test or inform it ("developed to back up the above hypothesis"). Please clarify how cell polarity is not directly coupled to chemotaxis (2 of the 4 facets of the model). And what about cell-ECM adhesion in terms of anchoring/traction? The presence and density of specific ECM ligands and their respective cell receptors (integrins) on the cell modulates cell-ECM interactions also, in addition to the ECM resistance or barrier function described here.

Response 3-2

As Reviewer 3 pointed out, some cells such as endothelial cells have clear front-rear polarity, and chemotaxis is directly associated with cell polarity. In NHBE cell migration, the effects of chemotaxis

is more prominent when there is a large number of cells and secretion ratio of activator and inhibitor is quite small. Therefore, the model in this paper was designed based on a large population of cells, and the polarity takes into account continuous cell movement, while chemotaxis is determined by the distribution of a large number of cells. Under this low resolution for individual cells, it is easier to calculate polarity and chemotaxis separately. In addition, the relation between cell polarity and morphogen secretion has not been determined yet in lung epithelial cells. In order to clarify the conditions, we have added the explanation of the conditions of the individual terms in the math section of the supplemental document.

Regarding cell-ECM adhesion, it is quite an important aspect not only as a physical but also a chemical interaction, as Reviewer 3 pointed out. The cell-ECM adhesion force could be another resistance force affecting cell migration, but as the Matrigel is covered on the top of the cell layer, the applied resistance force by cell-ECM adhesion is assumed to be consistent in the presented experimental assay. In that sense, the effects of cell-ECM adhesion on cell directionality should account for less than the other phenomena considered in the model. In order to simplify the calculations, we have omitted the effect of cell-ECM in this paper. We have added the above explanation in the math section of supplemental document.

3-3

On a related topic, later in the discussion ECM stiffness is considered but only in the sense that it is suggested that less stiff matrix would offer less resistance to migration. However, matrices have a diverse range of stiffness that is optimized for function. In fact, matrix that is too soft would not support migration at all because the traction forces could not be generated at adhesion sites.

Response 3-3

We agree with the point that Reviewer 3 raised. Cells migrate towards a lower resistance force within the appropriate range of ECM stiffness, but the multi-cellular organization is orchestrated by a diverse composition of cell and ECM, and the optimal stiffness depends not only on the cell type and organ function, but also on the developmental stage of the tissue. Therefore, soft and low resistance ECM is not always suitable for cells to migrate. We have added the above explanation in the discussion in page 12.

3-4

In the figure 6 experiments and modeling, why was a large area cleared instead of making single bead tracks which would more realistically mimic the action of cells on the matrix? Also discuss how did the lack of fibronectin deposition (as would be done by cell action) in the artificially remodeled space affect cells migrating into the area?

Response 3-4

One of the purposes of this paper is to show the importance of designing the cellular environment in order to reconstitute tissue pattern formation. As Reviewer 3 pointed out, a single bead track would mimic single cell movement more realistically. However, in order to reflect collective cell migration, more than one bead track is required, which cannot be made by optical tweezers within a feasible timeframe due to the very high amount of output force that would be required to do so. Additionally, as the effects of chemotaxis is only apparent with a large number of NHBE cells, the assay involving a large number of microbeads to artificially remodel the ECM in a relatively large area is more suitable to illustrate the purpose of the paper. The above explanation was added in page 9 to clarify the objective to employing magnetic force.

The experiments shown in Figure 6 do not contain added artificial FN, as the reviewer pointed out. In spite of the lack of FN, the result that collective cells progressively invaded into the degraded ECM area, in contrast to when cells could not migrate outside of the circular shape at the beginning without artificial degradation, indicates that the effects of lack of FN site contributes less to hinder cell migration compared to the resistance force from ECM. In other words, both the surrounding ECM and lack of FN site prevented cells from migrating to new sites, but cells can easily cross the FN boundary when pushed out by the many surrounding cells as shown in Fig. 1f, whereas cells cannot overcome the resistance force from ECM until it is weakened enough. The above discussion was added in page 9.

3-5

Finally, there is heavy, almost excessive, self-citation.

Response 3-5

As Reviewer 3 pointed out, some of the citations were duplicated. We have double-checked the citation list, and selected suitable ones for readers' better understanding.

REVIEWERS' COMMENTS:

Reviewer #1 (Remarks to the Author):

The authors have addressed all my previous comments to a satisfactory level. I recommend the work in its current form for publication.

Reviewer #2 (Remarks to the Author):

The authors have made significant efforts in addressing the comments raised by all of the reviewers. I particularly appreciate the detailed explanation on the rationale of cell-ECM boundary modeling and interface instability. The paper can be accepted for publication in its current form.

Reviewer #3 (Remarks to the Author):

This manuscript presents an informative investigation of the physical interactions in ECM remodeling and collective epithelial migration using innovative in vitro methods. These findings may form the basis for approaches to design or modulate cellular environments that direct tissue pattern formation. nearly all Reviewer comments have been addressed satisfactorily.

Weakening of resistance force by cell-ECM interactions regulates cell migration directionality and pattern formation

by Masaya Hagiwara, Hisataka Maruyama, Masakazu Akiyama, Isabel Koh, Fumihito Arai
Manuscript No. COMMSBIO-21-0550A

We are delighted to hear all reviewers have accepted our paper for the publication in Communications Biology. We thank all reviewers for thoughtful suggestions and insights, which have enriched the manuscript and produced a more balanced and better account of our research.

Reviewer #1 (Remarks to the Author):

The authors have addressed all my previous comments to a satisfactory level.
I recommend the work in its current form for publication.

Reviewer #2 (Remarks to the Author):

The authors have made significant efforts in addressing the comments raised by all of the reviewers. I particularly appreciate the detailed explanation on the rationale of cell-ECM boundary modeling and interface instability. The paper can be accepted for publication in its current form.

Reviewer #3 (Remarks to the Author):

This manuscript presents an informative investigation of the physical interactions in ECM remodeling and collective epithelial migration using innovative in vitro methods. These findings may form the basis for approaches to design or modulate cellular environments that direct tissue pattern formation. nearly all Reviewer comments have been addressed satisfactorily..